# Advances in Computational Approaches for Estimating Passive Permeability in Drug Discovery

**DOI:** 10.3390/membranes13110851

**Published:** 2023-10-25

**Authors:** Austen Bernardi, W. F. Drew Bennett, Stewart He, Derek Jones, Dan Kirshner, Brian J. Bennion, Timothy S. Carpenter

**Affiliations:** Lawrence Livermore National Laboratory, Livermore, CA 94550, USA; bernardi1@llnl.gov (A.B.); bennett69@llnl.gov (W.F.D.B.); he6@llnl.gov (S.H.); jones289@llnl.gov (D.J.); kirshner1@llnl.gov (D.K.); bennion1@llnl.gov (B.J.B.)

**Keywords:** passive permeability, biomembrane, molecular dynamics, machine learning, lipophilicity

## Abstract

Passive permeation of cellular membranes is a key feature of many therapeutics. The relevance of passive permeability spans all biological systems as they all employ biomembranes for compartmentalization. A variety of computational techniques are currently utilized and under active development to facilitate the characterization of passive permeability. These methods include lipophilicity relations, molecular dynamics simulations, and machine learning, which vary in accuracy, complexity, and computational cost. This review briefly introduces the underlying theories, such as the prominent inhomogeneous solubility diffusion model, and covers a number of recent applications. Various machine-learning applications, which have demonstrated good potential for high-volume, data-driven permeability predictions, are also discussed. Due to the confluence of novel computational methods and next-generation exascale computers, we anticipate an exciting future for computationally driven permeability predictions.

## 1. Introduction

The transport of small molecules such as drugs, toxins, and nutrients to and from various biological subsystems within living things is generally mediated through lipid membranes [1]. This includes transport in the skin [2,3], the lungs [4], the placenta [5], the intestines [6], the renal system [7], and the brain across the “blood brain barrier” (BBB) [8,9,10]. Interested readers are directed to Di et al. [11], which provides a general, comprehensive review of the biological aspects of passive permeability in drug design. Transport across biological lipid membranes may be characterized as either passive [11], where transport occurs mainly via diffusion across the membrane, or active [12], where transport is actively facilitated through transmembrane proteins. Although there are instances where membrane proteins may enhance the passage of small molecules, for the purposes of this work, we will refer to ‘passive permeability’ strictly as the unfacilitated, diffusive transport of a permeant across the lipids of a biomembrane. With the recent and rapid advancements in the field of drug discovery, developing methods for drug design and delivery that properly exploit passive permeability is often a critical task.

A therapeutic compound might be extremely potent, but if it cannot cross the cell membrane to reach the desired intracellular target, then the compound is ultimately not viable. Thus, understanding and incorporating the passive permeability of a compound into the drug discovery and development process is indispensable. To measure passive permeability for small molecules, in vitro methods such as PAMPA (Parallel Artificial Membrane Permeability Assay) [13,14] or various combinations of transcellular assays [15] are often used. However, these methods require time-consuming synthesis, purification, assay, and detection protocols that may require refinement for each new compound tested. Furthermore, certain permeability measurements (such as crossing the BBB) often necessitate in vivo measurements [16] that are arduous and expensive. Readers are again referred to Di et al. [11] for a recent review showcasing experimental studies concerning passive permeability. Computational methods to predict the passive permeability of compounds can augment experiments to aid and accelerate drug development [15].

The computational determination of passive permeability of small molecules across biomembranes can be categorized into three methodologies: lipophilicity relations, molecular dynamics (MD) simulations, and machine learning. Figure 1 shows an information flow diagram overviewing these categories.

Lipophilicity relations generally operate by generating a mathematical expression of the logarithm of a molecule’s partition coefficient between n-octanol and water (log *P_ow_*) as a function of the underlying molecular structure. Note that log *K_ow_* is another frequently used term for log *P_ow_* in the literature. The molecular structure is often represented by molecular descriptors, and the parameters of the model can be regressed against experimental data to improve predictive accuracy. Much of the foundational methodology developed for predicting passive permeability is laid out by Lipinski et al. [17], which notably outlines the “rule of five.” Early methods focused on characterizing lipid membrane permeability analogously via log *P_ow_* [18] since passive transport across a lipid membrane can be characterized as diffusion between hydrophilic (solution) and hydrophobic (membrane) environments. Log *P_ow_* is a primary component of the larger standard characterization of a therapeutic’s absorption, distribution, metabolism, excretion, and toxicity, or ADMET, properties [19,20]. More recent and sophisticated methods that rely on molecular simulation for estimating passive permeabilities have also been developed. These methods address significantly more of the complexity associated with actual biological systems to improve predictive capabilities over lipophilicity relations.

MD simulation has entered the spotlight over the past few decades, enabling high-resolution simulations of nano-scale systems; these simulations can be analyzed to provide direct permeability estimates [21,22,23]. Readers are directed to the comprehensive review by Venable et al. [21] of current passive permeability theory and studies using MD. MD simulation allows for explicit representation of much of the complexity of biological systems, such as ionic concentrations, membrane compositions, membrane additives, and anisotropic leaflets. While the in silico total sampling time of an atomistic MD simulation is generally restricted from nano- to microseconds [24,25], enhanced sampling techniques [26] provide an accelerated means to calculate free-energy profiles. Free-energy profiles can be used to estimate observables such as passive permeability, avoiding the computationally expensive sampling of explicit permeation events.

Characterizing the passive permeability of membranes with MD has generally been accomplished using one of three techniques: the homogeneous Solubility–Diffusion model (HSD); the Inhomogeneous Solubility–Diffusion model (ISD); or permeant counting methods. Depending on various model system features, each model retains appropriate use cases in current research applications.

Using Fickian diffusion theory, the HSD model mathematically expresses Overton’s Rule as
(1)$$P=\frac{K^{eff}D}{h},$$
where *P* is the estimated membrane’s permeability, $K^{eff}$ is the effective partition coefficient between the membrane and solvent, and *D* is the molecule’s diffusivity inside the membrane. Note that $K^{eff}$ can be straightforwardly approximated using $K^{eff}\approx P_{ow}$. It is also important to note that *P* and *P_ow_* and their respective logarithms represent related but separate quantities; it is somewhat unfortunate that ‘*P*’ has been widely adopted for both variables. Readers must take care to contextually identify whether a permeability or partition coefficient is being referenced in the literature, as general methods for determining log *P_ow_* are simply denoted by “log *P*” [17,27,28,29]. The HSD model provides a useful means to reconcile experimental log *P_ow_* data and actual membrane permeability estimates. Note that the HSD model cannot explicitly account for the heterogeneous nature of membrane permeation, an inherent limitation of the model. Comparing to log *P_ow_* often has the advantage of having more accurate and abundant experimental estimates. Recent Statistical Assessment of Modeling of Proteins and Ligands (SAMPL) challenges have focused on the blind prediction of log *P_ow_* and related quantities using many different computational approaches, allowing for a systematic comparison of methods [30].

The ISD model is derived from the Smoluchowski equation [31], which ultimately yields
(2)$$\frac{1}{P}=e^{-\beta F_{ref}}\int_{-h/2}^{h/2}\frac{1}{e^{-\beta F\left(z\right)}D_{⊥}\left(z\right)}dz,$$
where β is the reciprocal of the thermodynamic temperature, $F_{ref}$ is the reference free energy of the molecule in the surrounding solution, h is the membrane thickness, F(z) is the free-energy profile across the membrane, and $D_{⊥}\left(z\right)$ is the local diffusivity normal to the membrane. Using ISD models is a powerful and popular approach to quantifying passive permeabilities and is performed with umbrella sampling [32] using the weighted histogram analysis method [33] or other free-energy sampling methods [26]. The perpendicular membrane diffusivity can also be directly estimated from MD simulations [34].

Permeant counting methods employ flux or transition-based relations derived from Fickian diffusion theory. There are a variety of ways to estimate permeability from permeant counts; the reader is again referred to the comprehensive review by Venable et al. for underlying equations and details [21]. Permeant counting methods are best suited for small molecules with high permeability.

Machine-learning (ML) methods are gaining prominence within the broader field of cheminformatics. The reader is referred to Lo et al. [35] for a comprehensive review of current applications of ML to cheminformatics and drug discovery. The field of drug discovery has experienced significant growth in interest in the use of ML to predict experimental observables such as binding constants, lipophilicity, and passive permeability [36,37,38]. ML allows for the automatic incorporation of nonintuitive heuristics that further improve permeability estimates [39,40]. These heuristics can be built in a supervised setting where the relationship between features and labels (predicted values) are learned by the model or in an unsupervised setting where only relationships between unlabeled, featurized data are learned. Purely structural features such as extended-connectivity fingerprints (ECFP) make few assumptions about what is useful to the ML model but run the risk of poor generalization on unseen structures [41]. Features that are a set of calculated properties, such as ADMET properties, can provide higher-level information but do not explicitly contain information concerning the 3D structure. The ideal set of features is calculable for every sample in the largest available dataset, concisely represents the data, and does not omit any relevant information. Data availability influences feature selection, which in turn influences ML method selection. A prominent platform for making these decisions for molecular data is MoleculeNet, which contains a variety of ML methods and featurizations that have been tested on a panel of experimental datasets, including lipophilicity and blood–brain barrier penetration [38]. Depending on the availability of unlabeled or labeled data, both unsupervised and supervised methods have been successfully applied to studying passive permeability [42,43].

## 2. Prediction of Passive Permeability Using Lipophilicity Relations

Numerous computational methods using lipophilicity have been developed to quickly characterize a drug’s ADMET properties. While these methods have a reduced upper limit on accuracy, their speed is often invaluable, permitting nearly instantaneous predictive capabilities for any conceivable drug. These methods are generally used to perform initial drug candidate screening, after which more sophisticated analyses may be used to refine candidate selection. Both commercial (MOE www.chemcomp.com (accessed on 19 September 2023), Schrödinger www.schrodinger.com (accessed on 19 September 2023), Dragon [44]) and open-access [45,46,47,48] methods and software have been developed to permit rapid ADMET calculations. These software packages utilize the lipophilicity characteristic log *P_ow_*, the logarithm of the *n*-octanol and water partition coefficient, which can be correlated to passive permeability via “Overton’s Rule”. A diverse set of log *P_ow_* models are actively employed in current ADMET software [17,27,28,29,49].

Another set of log *P_ow_* methods uses quantum calculations and implicit solvation models. For example, the COSMO-RS method has been used for predicting small molecule partitioning [50]. A recent comparison to atomistic MD simulations showed that COSMO-RS provides relatively high-accuracy log *P_ow_* predictions when compared to experimental data [51].

A molecule’s estimated log *P_ow_* is often used in machine-learning studies due to its straightforward and rapid calculability [42,43,52,53,54,55,56]. Using log *P_ow_* as a feature is attractive for both simpler models, such as random forests, and more complex models, such as neural networks. Log *P_ow_* has been successfully implemented in machine-learning studies for predicting standard membrane drug permeabilities [42], blood–brain barrier permeability [43,53,54], and permeability-associated properties such as export inhibition constants [55].

While log *P_ow_*-based permeability models provide a useful ability to rapidly characterize lipophilicity in drug discovery, their simplicity stands as an inherent limitation for quantifying passive permeability for more complex systems, such as drugs with complex chemical features or heterogeneous membrane environments. There are often assumptions made that the molecule being assessed contains chemical moieties that are adequately represented within the experimental data from which the prediction models were constructed. For complex systems requiring more accurate estimates of passive permeability, more sophisticated methods are required.

## 3. Passive Permeability Studies Using Atomistic Molecular Dynamics

Sampling systems with MD simulations produce time-evolved trajectories at atomistic resolution. MD simulations can be directly applied to studying passive permeability when combined with permeability theory [21]. MD simulations have the potential to enable much greater fidelity than the well-established lipophilicity relations since MD can explicitly model complex system features, such as heterogeneous membrane compositions and nano-scale interfacial effects [21]. Classical MD can also be further enhanced with more intricate sampling techniques, such as constant-pH simulations [57], polarizable MD [58], or ab initio MD [59]. However, it is worth noting that as the methodological complexity of sampling increases, the computation cost and opportunities for incidental systematic errors also increase.

### 3.1. Inhomogeneous Solubility-Diffusion

ISD approaches have been robustly implemented during the last decade [60,61,62,63,64,65,66,67]. While ISD has several validated successes at predicting passive permeability, it can be computationally intensive and is contingent on the accuracy and reliability of underlying collective variables and parameters.

Sugita M. et al., 2021 [63] performed a comprehensive study computing passive permeabilities of 156 cyclic peptides using ISD. Only one protonation state for each side chain was considered. Also, only cyclic peptides with an AlogP value less than 4.0 were modeled, as the experimental results for those with values greater than 4.0 are thought to be unreliable due to low peptide solubility [68]. Permeabilities and energy barriers were quantified and compared to experimental and AlogP data. Performance was assessed with standard statistical metrics and generally yielded middling results in precision and accuracy. This study exemplifies that, even with restraints applied to simplify the quantification of permeability using ISD, accurate quantification and relation to experiments remain a challenge.

Yue Z. et al., 2019 [64] studied pH-dependent membrane permeation of propanolol using continuous constant-pH MD (CPMD), which enables smooth dynamic protonation of molecules during simulation. CPMD uses a continuous variable ranging from zero to one that represents the (non-physical) partial protonation state, combined with a hybrid explicit/implicit solvent model. The permeability calculated from CPMD was found to be similar in value to the permeability calculated from the minimum of the PMFs of the individual protonated states (without CPMD). The orientational effects on permeation were also studied using 2D PMFs. This study showcases the latest computational techniques to study membrane permeation using dynamic protonation in detail.

### 3.2. Permeant Counting Studies

Permeant counting methods calculate passive membrane permeability using unbiased simulations that explicitly count permeant transport events [69,70,71,72,73]. These methods are generally best applied to high-permeability permeants since the methods’ accuracy is directly related to the number of permeation events sampled during the simulation. Permeant counting methods are attractive since they do not require any artificial bias to be applied to the simulated system. However, this can also be detrimental to sampling since artificial biases accelerate the sampling of conformational space. Due to these sampling concerns, studies using these methods often also provide comparisons to ISD results.

Mahdi G. et al., 2020 [70] studied concentration-dependent passive permeation of ethanol using flux-based and transition-based permeant counting methods and compared these to a Bayesian inference ISD method. Interestingly, with respect to experimentally calculated permeabilities, it was found that permeabilities were significantly overestimated using the ISD method, while permeabilities were overestimated to a lesser degree using permeant counting methods. Various possible explanations for these discrepancies are provided, such as the unstirred layer effect [74] and force-field inaccuracy.

Krämer A. et al., 2020 [71] studied permeabilities of oxygen, water, and ethanol using permeant counting and also compared to Bayesian-inferenced ISD. They were able to obtain more precise results using permeant counting compared to ISD, while there were issues with Markovian sampling at the membrane-water interface for permeant counting. In general, the authors note that standard additive force fields for these molecules are generally insufficient to yield agreement with experimental results, and polarizable force fields might be required. Of particular note, the open-source packages RickFlow (http://www.gitlab.com/olllom/rickflow (accessed on 19 September 2023)) and diffusioncma (http://www.gitlab.com/olllom/diffusioncma (accessed on 19 September 2023)) are introduced and made available in their paper, which provide means to facilitate future permeability simulations and analysis, respectively.

## 4. Applications Using Coarse-Grained Molecular Dynamics

The high computational cost of atomistic simulations has motivated the development and utilization of coarse-grained (CG) models. Consequently, there have been a large number of passive permeability studies that have employed CG MD [75,76,77,78,79,80,81]. CG models reduce the degrees of freedom of the simulation, enabling larger simulations with longer timesteps and greater computational efficiency. These sampling enhancements come at the cost of atomic resolution detail, accuracy, and dynamics. Due to the expense of these trade-offs, it is especially important to employ experimental comparison for CG MD studies. The popular Martini model, which roughly maps three to four heavy atoms to a single interaction site or “bead”, was parameterized to reproduce the free energies of transfer between water and organic liquids for a variety of chemical building blocks [82]. The speed of the Martini model allows for higher energy barriers to be crossed with relatively low computational cost, allowing permeation to be directly observed in silico, such as for cholesterol [83]. It is also possible to simulate larger molecules permeating membranes with the Martini model, with notable examples including antimicrobial peptides [84], gold nano-particles [85], and DNA-encapsulated nano-particles [86]. The large number of parameters available in the Martini force field also allows simulations of large and complex lipid mixtures, such as realistic bacterial membranes [87], viral membranes [88], and plasma membrane models [89].

In a series of works, Bereau and co-workers ran extensive MD campaigns for Martini small molecules crossing lipid membranes [76,78,90]. ML models were then built on these calculations, extrapolating the potential chemical space covered within the model [42]. Beyond the current scope of this article, we note extensive efforts to use ML modeling to generate CG models, which can be further employed for multiscale modeling [90,91,92,93,94]. For example, recently, Li et al. used multiscale modeling with ML to convert CG lipid membrane structures to atomistic detail [94].

## 5. Applications of Machine Learning

Traditional machine-learning or non-deep-learning methods have been applied to various problem domains, including molecular information and modeling, with no exception to passive permeability [35,43,53,54,55,95]. These include traditional ML models such as support vector machines (SVMs) as well as k-nearest neighbors (KNN). Deep-learning techniques, which involve some form of sophisticated neural network, have recently become a key subject of computational research concerning lipophilicity and, by extension, passive permeability [42,52,55,56,96,97]. With increasingly larger collections of available data in recent years, both labeled and unlabeled, as well as advancements in data processing capabilities, the adoption of ML techniques has increased at a seemingly exponential pace. MoleculeNet, a prominent benchmark that aggregates numerous molecular datasets and ML methods, predicts various molecular descriptors, including passive permeability [38]. MoleculeNet provides useful benchmark comparisons to compare state-of-the-art ML methods to standard, validated methods. Additionally, MoleculeNet aggregates a representative set of not only traditional ML methods but state-of-the-art deep-learning techniques such as graph convolution neural networks (GNNs) across 17 distinct molecular datasets.

There are a number of recent studies that combine ML models with MD simulations to study small molecule permeation [39,40,42,43,52,98]. Riniker and co-workers have developed molecular dynamics fingerprints (MDFPs), where short MD simulations of a molecule were performed and analyzed to give time-averaged molecular features that improved ML predictions [98]. Recently, high-throughput atomistic MD free energy calculations for small-molecule transfer from water to cyclo-hexane were used to train different ML models [52]. After training, the ML model had a similar error in predicting the MD free energy compared to the average MD sampling error for new molecules. The computational cost of the ML model was a fraction of running the MD free energies; training on thousands of small molecules could lead to predictions for millions.

Another active area of research is using ML methods to analyze trajectories from MD simulations, which is an attractive application for ML, given the multi-dimensional nature of molecular simulation data. Unsupervised approaches can learn a latent representation of a reaction that allows scientists to better identify rare or interesting states [99]. Other ML-based methods that isolate collective variables are also under development, with the potential for application to membrane permeation [100,101].

Additionally, neural network-based potential energy functions (NNPs) have demonstrated the possibility of extrapolating the more expensive quantum mechanical (QM) potential energy functions. NNPs show competitive accuracy when compared to other popular approximation methods [102]. Simultaneously, open-source tools are being developed using the popular deep-learning framework PyTorch to aid in the development of novel ML force fields that can be trained with backpropagation [103,104].

## 6. Current Limitations and Outlook

Predicting the localization and distribution of small molecules in biological systems remains critically important for drug discovery and the general understanding of natural biological processes. These predictions depend on the reliable determination of passive permeability in biological membranes for small molecules, which is experimentally expensive. Consequently, the development and utilization of computational methods to predict passive permeability have been researched extensively, and we anticipate further growth in the future. In addition to providing accurate predictions for membrane permeation for drug discovery platforms, simulations have provided key mechanistic insights into membrane biophysics.

While there has been great progress, improvements to current computational methods are needed to accurately and efficiently estimate permeability. Research competitions, such as the SAMPL challenges, are a key driver for novel methodology development and validation [30,105]. The following sections discuss key areas for improvement, which motivate future research: force-field development and small-molecule parameterization, computational sampling, experimental comparison, and machine learning. Finally, a concluding long-term outlook is provided.

### 6.1. Force-Field Development and Small-Molecule Parameterization

A number of classical, atomistic force fields have been developed in recent decades, which have been heavily utilized for passive permeability simulations. The most widely used force fields include AMBER [106,107], GROMOS [108], and CHARMM [109]. However, classical force fields have inherent limitations, such as lacking polarizability and dynamic bonding, and more complex models will be needed to fully characterize these more complex chemical interactions. A CHARMM polarizable force field [110] is under active development and is now available for simulations, which can permit improved accuracy for systems that require polarizable interactions. There has also been progress for constant-pH force-field implementations [111], which is particularly valuable for permeability simulations that contain protonatable groups. Force fields that universally implement dynamic bonding are currently not efficient enough to meaningfully contribute to passive permeability MD simulations, but likely represent the long-term future of the field. Fast ML based QM methods, such as the ANI-potential, will make large dynamically bonded simulations possible in the future. Expanding the complexity of biological model membranes and improving lipid force fields will continue to remain active areas of research.

There has also been progress in CG force-field development, which enables even greater computational efficiency, albeit with a significant cost of accuracy. Martini 3 has recently been released, which provides the ability to model more specific chemical interactions [112]. ML-derived CG force fields are also being developed, which can improve CG accuracy and help automate parameterization. While CG force fields have limited accuracy, their ability to reliably reach sampling equilibrium has proven highly useful for permeability studies [75,76,77,78,79,80,81] and will remain an integral component of passive permeability simulations for the foreseeable future.

Once a force field is defined, generating a molecular topology for an unparameterized small molecule can be a large computational and human-intensive task. Software that automate molecular topology building have been developed and widely utilized. Automated topology builders include GAFF for AMBER topologies [113], cgenFF for CHARMM-compatible parameters [114,115], and the Automated Topology Builder (ATB) (https://atb.uq.edu.au (accessed on 19 September 2023)) [116] for GROMOS molecules. For CG systems, an automated small-molecule topology builder has recently been developed [117]. In general, care must be taken when using automated topology generators, as the accuracy of resultant parameters can vary significantly. We anticipate many novel topology-generation methods in the near future, with ML-assisted frameworks being actively explored.

### 6.2. Computational Sampling

The predictive capability of MD simulations is often practically limited by the amount of computational sampling. The advent of GPU computing has significantly increased feasible sampling windows, providing up to an order of magnitude speedup over traditional CPU-based MD simulations [118]. CPUs are typically limited to around one hundred parallel cores for high-performance architectures, while GPUs can support thousands of simultaneous simple computations, which is well suited to the nature of non-bonded interactions in MD simulations. The algorithmic development and standardization of GPU computing has promising potential for significant performance improvements in the future and poses a fundamental challenge in current computational research [25].

Improving computational sampling in MD simulations is another active and important area of methodological research. More efficient free-energy calculations are central to improving permeability predictions for ISD models, which has been a long-standing challenge rooted in statistical mechanics. Free energy is difficult to calculate, largely due to extensive sampling requirements for convergence of the entropic component. One important consideration for membrane permeation simulations is that lipid bilayers are soft, fluid-phase systems that can deform to accommodate a molecule. Careful selection of an appropriate reaction coordinate and ensuring sampling convergence are critical tasks. Additionally, the permeant’s chemical structure may cause challenging sampling problems, especially for long, flexible molecules or charged molecules. Other considerations can further complicate predicting the permeability of charged molecules, including charge neutralization, polarizability, and protonation state. More sophisticated enhanced sampling methods are also being developed to overcome these limitations. The development of simulation methods that can efficiently accommodate multiple reaction coordinates will be useful in the future, especially for modeling aggregative permeation and for modeling molecules that are commensurate in size to the bilayer. Recently, a number of enhanced sampling methods have been applied to membrane permeation, including weighted ensemble [119], metadynamics [120], and replica-exchange umbrella sampling [121]. Multiscale methods for computing small molecule permeation across lipid bilayers are also being developed [122]. The development and utilization of novel enhanced sampling techniques will increase the relevance of MD simulations for large-scale drug-discovery screens.

### 6.3. Experimental Comparison

Passive permeability estimates for small molecules across biomembranes can be independently obtained from both real-world experiments and computer simulations, enabling straightforward comparisons. However, ensuring both models are compositionally similar and consistent is often challenging. Prominent techniques for experimentally determining permeability include Caco-2, PAMPA, and other cellular-based assays [11]. Data from commonly used in vitro PAMPA experiments can be inconsistent, as some assays contain lipid bilayers, while others are lipid–oil–lipid tri-layers [14]. While the tri-layer model reduces solvent-induced permeability, certain highly hydrophobic compounds can be mischaracterized. Specifically, compounds that contain intermediate-length alkyl “tails” appear to have an increased affinity to the oil that exists between the lipid layers. Retention of compounds in this oil will not provide any useful data as no detectable compound will be present in the secondary well. MD simulations can add context to such instances by showing a very favorable energy at the center of a lipid bilayer [23,122]. Studies that unify experimental and simulation passive permeability data are critical for advancing both areas of research.

### 6.4. Machine Learning

Machine learning has broadly entered the space of computational drug design, and permeability is no exception. In recent years, due to the proliferation of GPUs, deep learning has quickly gained traction as a fruitful area of research in molecular modeling generally. Deep learning allows the model to automatically learn task-specific “features” from the data that can, in many cases, outperform the “expert-knowledge” features that have traditionally served as inputs to traditional machine-learning models. However, deep learning in this field generally suffers from a lack of experimental data. Databases such as ChEMBL [123] have made significant progress in the space of data aggregation and curation [38], but dataset fidelity is still a significant issue, and orders of magnitude more chemical data remain unutilized and uncollected. MD simulations can be used to supplement molecular data needs, but accuracy and sampling issues discussed in Section 6.1 and Section 6.2 drive the need for experimental validation.

MD trajectory data are relatively accessible and contain structural and dynamic details that are often missing from experimental datasets, such as time-resolved hydrogen bonding, root-mean-square deviation, and solvent-accessible surface area, but their featurization is an open question in the research community. One promising research focus in this direction is to develop models that can exploit the structural properties of chemical data by representing small molecules as point clouds or spatial graphs [124,125,126,127,128,129]. Several software tools that utilize point clouds are available and under active development [130,131,132]. PyTorch Geometric has greatly enabled the training and development of graph-based neural networks [133], and TorchDrug has additionally provided software infrastructure for the application of these methods to molecular datasets [131]. These tools lend themselves to improving the analysis of three-dimensional MD trajectory data.

Additionally, the learning of molecular representations in the presence of large collections of unlabeled or sparsely labeled datasets will continue to be a significant focus of deep-learning research as traditional fingerprinting methods have significant limitations [127,134,135,136]. The ChemBERTa language model uses large amounts of unlabeled, string-based molecular encodings as inputs and uses self-supervised methods to train an initial model, which is then fine-tuned using labeled data [137]. Better usage of unlabeled and labeled data for ML models suggests a bright future for machine learning for permeability.

### 6.5. Long-Term Outlook

We anticipate an exciting and productive future for computational chemistry, particularly in drug discovery and membrane permeation. In the next few years, more exascale machines will be online, yielding order-of-magnitude increases in computational power, allowing for more molecules to be studied with more accurate calculations. Future studies, including the influence of transmembrane transporters and channels for predicting total membrane permeability, will provide even more biologically relevant predictions. The growth of new algorithms and the incorporation of ML into many aspects of computational chemistry will continue. Linking ML results to human-interpretable explanations will remain a crucial challenge, along with experimental data validation.

## Figures and Tables

**Figure 1 membranes-13-00851-f001:**
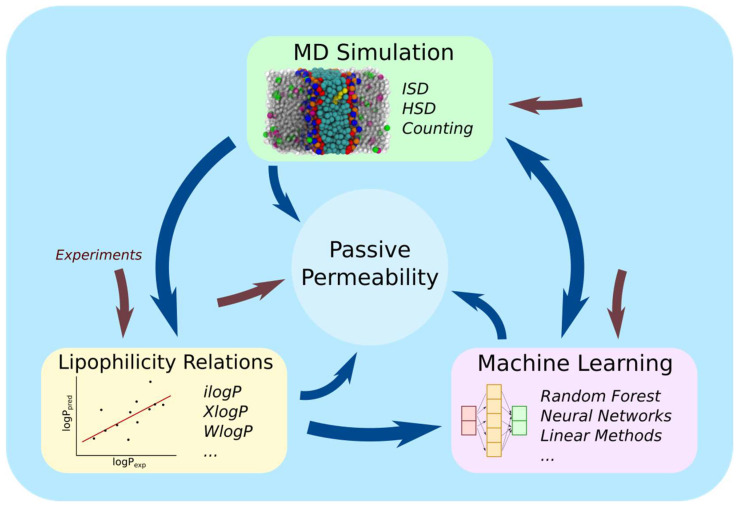
Information flow diagram of the different methodologies for computationally estimating passive permeability for biomembranes. Directed arrows indicate information flow. Experimental data are represented by brown arrows. ISD stands for the inhomogeneous solubility diffusion methods, while HSD stands for homogeneous solubility diffusion methods.

## Data Availability

Not applicable.

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
