# Peer review of "Advances in Computational Approaches for Estimating Passive Permeability in Drug Discovery"

_membranes, 2023, doi:10.3390/membranes13110851_

Round 1

Reviewer 1 Report

This manuscript describes very clearly and provides examples of protocols to predict the permeability of small and neutral molecules. For the sake of clarity please be careful and describe that is for small and neutral molecules. there are examples such as the transport of fatty acid that is greatly facilitated by membrane proteins,  prostaglandin E2 that permeate connexin43 hemichannels,  and several others under physiological cross the lipid bilayer of the cell membrane via transporter or non-selective membrane channels. Otherwise is a well-organized contribution.

Reviewer 2 Report

The review "Advances in Computational Approaches for Estimating Passive Permeability in Drug Discovery" presented by Bernardi et al. resumes as stated in the title the current state of the art of computational methods to simulate and predict membrane permeability of novel drugs. Authors give a brief introduction on the underlying theories, then continue and introduce different computational methods and finish with a brief outlook where they stress the high potential of this type of methodology.

The ms is very interesting because it allows, especially the general reader, to understand that a combination of different computational methods is necessary to optimize the experimental "wetlab" design and that these approaches have the potential to reduce time and costs to obtain potential lead compounds.    However, I think that this format is not optimal, as it is too basic for the specialized reader and too superficial for the general reader. I would be better to focus on one group. This will allow the author to be more specific. 

I also miss a discussion on the feedback to "real" experiments. Authors should showcase at least one example. 

specific comment:

line 87: ...often prohibitively expensive....

prohibitively is not scientific, thus this type of opinion should either be avoided of stated in a different way. 

Reviewer 3 Report

The review deals with passive permeation of cellular membranes and enumerates a variety of computational techniques currently utilized to facilitate the characterization of passive permeability.  A chapter devoted to a description of various machine learning applications opens the way to new possibilities. This review provides a good summary of the recent developments in the field. The background and recent developments are reviewed, summarized, and highlighted adequately. The critical evaluation of the literature results, particularly the challenges and future directions in this research area could have been further improved.
